# Academic, clinical and personal experiences of undergraduate healthcare students during the COVID-19 pandemic: A prospective cohort study

Sonyia McFadden\*, Sharon Guille, Jean Daly-Lynn, Brenda O'Neill, Joanne Marley, Catherine Hanratty, Paul Shepherd, Lucia Ramsey, Cathal Breen, Orla Duffy, Andrea Jones, Daniel Kerr, Ciara Hughes

School of Health Sciences, Institute of Nursing and Health Research, Ulster University Jordanstown Campus, County Antrim, Northern Ireland

\* s.mcfadden@ulster.ac.uk

**Data Availability Statement:** All relevant data are within the paper and its Supporting Information files.

## Abstract

### Background

Coronavirus disease 2019 has impacted upon the role and safety of healthcare workers, with the potential to have a lasting effect on their wellbeing. Limited research has been conducted during previous pandemics exploring how student healthcare workers are impacted as they study and train for their professional careers.

### Objective

The aim of the current study was to examine the specific impact of COVID-19 on the academic, clinical and personal experiences of healthcare students.

### Method

Undergraduate students across three year groups within the School of Health Sciences at Ulster University completed online Qualtrics surveys at three timepoints during one academic year (2020/2021). Quantitative survey data was downloaded from Qualtrics into SPSS Version 25 for descriptive analysis of each cross-sectional sample. Qualitative survey data was downloaded into text format, which was thematically analysed using content analysis.

### Results

412 students completed the survey at Time 1 (October 2020), n = 309 at Time 2 (December 2020) and n = 259 at Time 3 (April 2021). Academically, the pandemic had mostly a negative impact on the learning environment, the development of practical skills, the assessment process and opportunities for peer learning and support. Students reported increased stress and challenges managing their workload and maintaining a sense of motivation and routine. Clinically, they felt unprepared by the university for placement where the pandemic had an increasingly negative impact over time on learning and skill development. In terms of

**Funding:** The authors received no specific funding for this work.

**Competing interests:** The authors have declared that no competing interests exist.

personal experiences, despite the majority of students taking steps to keep physically and mentally well, negative impacts on friendships, mental wellbeing and concerns for family were reported. The pandemic had not impacted upon career choice for most students.

## Conclusion

Consideration must be given to the development of practical skills so students feel prepared for their professional careers given the practical nature of their roles. Programme coordinators should adopt a holistic approach to student wellbeing.

## Introduction

In December 2019, a novel coronavirus disease emerged and in a matter of weeks Coronavirus disease 2019 (COVID-19) was designated a pandemic [1]. Subsequent impacts on the role and safety of healthcare professionals have been reported including exposure to high risks of infection, inadequate protection against contamination, overwork, frustration, discrimination, isolation, patients with negative emotions, exhaustion and a lack of contact with their families [2].

Healthcare students complete clinical placements to enhance learning alongside their academic studies and have been identified as positive contributors to the healthcare workforce during previous disasters or pandemics [3]. Forced changes to curriculum delivery and design in response to COVID-19 are likely to have a profound effect on student's learning experiences. Students are also likely to be exposed to increased demands and challenges in the clinical setting. Prior to the onset of the current pandemic, the use of digital technologies in higher education grew rapidly [4]. While accepted as being essential to the future of university education, uptake of the breadth of available technologies for learning and teaching purposes was varied [5] prior to the necessity to move fully online in March 2020. Methods previously used have included the use of Technology Enhanced Learning (TEL), Virtual Learning Environments (VLE) and online learning. This type of educational provision is flexible and readily available, offering students the opportunity to learn at their own pace in the comfort of their own homes, thereby reducing travel time and costs. Disadvantages of this method however have been reported including a sense of isolation, loss of personal interaction and a lack of face-to-face interaction [6].

Decreased interactions may also impact upon students' opportunities for peer learning and support. Peer learning encompasses a broad spectrum of activities with learners working in pairs or groups with varying levels of staff involvement. Within a clinical setting, peer learning may be used to help students integrate practical skills with theoretical knowledge. Systematic reviews of this type of learning in clinical education highlight its numerous positive effects [7, 8]. These include improved communication skills and team working abilities [9]; increased accountability and responsibility [10]; increased confidence and task efficiency [10, 11]; improved critical enquiry and reflection [9, 12]. Furthermore, peer learning is reported to facilitate a supportive clinical experience [11]; enhance socialization and improve transition into practice [9, 10]; and decrease student anxiety [11, 12]. As well as these benefits, it also affords students emotional support as part of the learning process. Friendships which develop consequent to peer learning are reported to facilitate a sense of community and support amongst the student cohort [13, 14]. Interactions between students and teachers is important for enhancing mental health and wellbeing [15] with the pandemic occurring at a time when there is already an increase in mental health issues in the general population and amongst the student cohort [16].

As adult learners, it is likely that students have responsibilities outside of academia such as employment, family and caring responsibilities that may also be impacted by COVID-19, which in turn may add increased burden to healthcare students in addition to the academic and clinical challenges they may face. Previous research conducted with medical and allied health students indicate concern for themselves or their family members becoming infected with the virus [17]. Although there is some research concerning the impacts and coping strategies of the current and previous pandemics on healthcare staff, there is limited reporting in the literature concerning how healthcare students cope and the overall impact on their experiences.

Therefore, the aim of this study was to examine the impact of COVID-19 on the academic, clinical and personal experiences of healthcare students. It is anticipated data from the study will contribute to understanding the specific impact of the pandemic on healthcare students as they study for professional careers. This information will be beneficial for identifying educational and clinical training needs to ensure optimal transition into the healthcare workforce.

## Methods

### Research design

A prospective cohort design was implemented to meet the aim of the study. The study was set in Ulster University Northern, which is geographically spread across 4 campuses within Northern Ireland. The University has interprofessional training incorporated across all health courses with placements carried out in local University teaching hospitals.

### Sample

The purposive sample included n = 892 undergraduate students from within the School of Health Sciences at Ulster University registered on one of the following courses: Podiatry, Diagnostic Radiography and Imaging, Radiotherapy and Oncology, Physiotherapy, Occupational Therapy, Health Science Physiology and Speech and Language Therapy. Inclusion criteria extended to students from all three year groups who could be on either academic or clinical placement. Ulster University students from outside the School of Health Sciences were excluded from the study.

### Measure

Ethical approval for the study was obtained from Ulster University Nursing and Health Research Ethics Filter Committee (FCNUR-20-020). Following a review of the literature and in collaboration with peer researchers as PPI representatives, an online survey was developed using Qualtrics software. Prior to the current study, a pilot survey was conducted with undergraduate healthcare students and changes to the final survey were made following a review of the feedback. The survey design included open and closed questions to enable the collection of quantitative and qualitative data for analysis. Questions explored healthcare student's academic, clinical and personal experience including: the method and quality of remote learning, the development of practical skills, the impact in the clinical setting, thoughts and feelings about the pandemic and the impact on support, wellbeing, family, finances and career choice.

### Data collection and analysis

Students were invited to participate via the internal university email system. The email invite included a participant information sheet and a link to access the anonymous online survey via Qualtrics. Consent was obtained at the beginning of each survey, the completion of which

were also taken as an indication of implied consent. Students completed surveys at three time-points across the academic year: Time 1 (T1) in October, Time 2 (T2) in December 2020 and Time 3 (T3) in April 2021. Quantitative survey data was downloaded from Qualtrics into SPSS Version 25 for analysis. A cross sectional analysis was used to generate descriptive statistics to provide a snapshot of student's experiences at focal timepoints. Some results included two or three timepoints to enable comparisons over time; other results were specific to each timepoint due to variations in the survey questions. Results are presented as valid percentages, excluding missing data where applicable. Qualitative data was also downloaded from Qualtrics into text format, which was then coded and thematically analysed using content analysis. Data from the survey was checked by the research group and consensus reached through discussion.

## Results

### Demographic information about the sample

The number of healthcare students who participated in the study across each timepoint were as follows: Time 1 (T1) in October 2020 (n = 412), Time 2 (T2) in December 2020 (n = 309) and Time 3 (T3) in April 2021 (n = 259). At T2 n = 261 (86.4%) had completed T1 and at T3 n = 211 (86.5%) had completed T1 and n = 199 (81.9%) had completed T2. Characteristics of the sample across all three timepoints are provided in "Table 1". The majority of the sample were female (80.6% T1, 83.9% T2, 85.2% T3). Respondents were aged less than 20 years old (35.5%, 29.3%, 31.1%), 20–23 years (43.6%, 47%, 43.4%), 24–30 years (11%, 11.2%, 11.9%) and over 30 years (9.8%, 12.5%, 13.5%) from T1 to T3. Participants represented all professions within the School of Health Sciences at Ulster University with responses by year group (1st, 2nd, 3rd) across all three timepoints as follows: 41.5%, 27%, 31.4% (T1), 32.6%, 30.9%, 36.5% (T2) and 45.9%, 33.2%, 20.9% (T3).

A smaller percentage of the sample had caring responsibilities (16.2%, 18.8%, 21.9%) and a larger percentage had a part time job alongside their studies (75%, 74.7%, 71.7%). Across all three surveys, approximately 30% of students reported having a term time address that differed to their home address. A minority of students reported living with someone who was shielding or high risk (25.6%, 23.3%, 22%) or altering where they would normally live to protect others (13.9%, 12.1%, 14%).

The impact of shielding on students was noted from a financial, social and psychological perspective. Whilst students provided comments on each of these strands individually, they were interconnected in some cases. Financial difficulties arose from not being able to go back to work, reducing hours to limit risk and increased costs associated with contractual agreements for accommodation, travel and living expenses. Some students reported financial benefits from furlough, being able to move home or work more. Social impacts centred around difficulties with not seeing friends and family with comments reflecting a proactive cautious approach taken by students in terms of choosing to avoid or limit social interactions. The benefit of this approach meant they knew they were protecting their loved one. Indeed, psychologically, a common concern voiced by students was the fear of transmitting COVID to a loved one. The lack of social interactions however impacted upon students psychologically as they expressed feelings of loneliness and isolation, increased stress and anxiety. For a few students, shielding did not have any impact.

> "*I was very nervous going and seeing people even when it was allowed as I didn't want to neg-atively affect those living with me...*" (Second Year, Physiotherapy undergraduate student, Timepoint 1)

**Table 1. Characteristics of the sample.**

| | | Time 1 | | Time 2 | | Time 3 | |
| --- | --- | --- | --- | --- | --- | --- | --- |
| | | N = 412 | | N = 309 | | N = 259 | |
| | | N | % | N | % | N | % |
| **Gender** | | | | | | | |
| Male | | 76 | 18.7 | 48 | 15.8 | 35 | 14.3 |
| Female | | 328 | 80.6 | 255 | 83.9 | 208 | 85.2 |
| Other | | 3 | 0.7 | 1 | 0.3 | 1 | 0.4 |
| **Total** | | **407** | **100** | **304** | **100** | **244** | **100** |
| **Age** | | | | | | | |
| Less than 20 years old | | 145 | 35.5 | 89 | 29.3 | 76 | 31.1 |
| 20–23 | | 178 | 43.6 | 143 | 47.0 | 106 | 43.4 |
| 24–30 | | 45 | 11.0 | 34 | 11.2 | 29 | 11.9 |
| More than 30 years old | | 40 | 9.8 | 38 | 12.5 | 33 | 13.5 |
| **Total** | | **408** | **100** | **304** | **100** | **244** | **100** |
| **Employed alongside course registration** | | | | | | | |
| Yes | | 306 | 75.0 | 227 | 74.7 | 170 | 71.7 |
| No | | 102 | 25.0 | 77 | 25.3 | 67 | 28.3 |
| **Total** | | **408** | **100** | **304** | **100** | **237** | **100** |
| **Caring Responsibilities** | | | | | | | |
| Yes | | 66 | 16.2 | 57 | 18.8 | 52 | 21.9 |
| No | | 342 | 83.8 | 247 | 81.2 | 185 | 78.1 |
| **Total** | | **408** | **100** | **304** | **100** | **237** | **100** |
| | **Course Registration** | | | | | | |
| Podiatry | | 24 | 5.9 | 22 | 7.2 | 15 | 6.1 |
| Diagnostic Radiography and Imaging | | 89 | 21.8 | 79 | 26 | 61 | 25 |
| Radiotherapy and Oncology | | 20 | 4.9 | 26 | 8.6 | 21 | 8.6 |
| Physiotherapy | | 99 | 24.3 | 42 | 13.8 | 33 | 13.5 |
| Occupational Therapy | | 75 | 18.4 | 87 | 28.6 | 83 | 34 |
| Health Science Physiology | | 48 | 11.8 | 30 | 9.9 | 16 | 6.6 |
| Speech and Language Therapy | | 53 | 13 | 18 | 5.9 | 15 | 6.1 |
| **Total** | | **408** | **100** | **304** | **100** | **244** | **100** |
| **Year of Registration** | | | | | | | |
| **First** | | **169** | **41.5** | **99** | **32.6** | **112** | **45.9** |
| **Second** | | **110** | **27.0** | **94** | **30.9** | **81** | **33.2** |
| **Third** | | **128** | **31.4** | **111** | **36.5** | **51** | **20.9** |
| **Total** | | **407** | **100** | **304** | **100** | **244** | **100** |

'*(I) Had to quit part time job as going on placement was increasing the risk of bringing COVID home enough without the addition of a part time job with the public"... (Third Year, Physiotherapy undergraduate student Timepoint 1)*

"*It has made me more aware of who I mix with and whether or not my interactions are necessary….*" (First Year, Radiotherapy and Oncology undergraduate student, Timepoint 1)

## Academic experiences

At the time of each survey, a greater number of students indicated that they were currently in university as opposed to clinical placement "Table 2". By T2 and T3, 42.1% and 74.2% of students respectively had been on campus mainly to attend practical lessons.

**Table 2. Current status of students.**

| | Time 1 N = 412 | | Time 2 N = 309 | | Time 3 N = 259 | |
|---|---|---|---|---|---|---|
| | N | % | N | % | N | % |
| Placement | 56 | 13.7 | 43 | 15.4 | 43 | 19.5 |
| University | 314 | 77 | 205 | 73.5 | 160 | 72.4 |
| Other | 22 | 5.4 | 21 | 7.5 | 17 | 7.7 |
| University and Other | 8 | 2 | 3 | 1.1 | | |
| University and Placement | 8 | 2 | 7 | 2.5 | 1 | 0.5 |
| **Total** | **408** | **100** | **279** | **100** | **221** | **100** |

Of importance to remote learning, the majority of students had access to a laptop suitable for academic work (92.2%, 94.4%, 92%). Less students however reported having access to reliable fast broadband (70.6%, 69.7%, 67.1%), a desk suitable for academic work (73.5%, 70.1%, 71.3%) and a quiet/adequately sized working space (68.4%, 68.1%, 71.3%). At T3, 12.2% reported computer literacy had impacted upon their ability to learn online.

The majority of students selected a preference for teaching that was delivered synchronously as opposed to asynchronously over T1 (76.9% vs 23.1%) and T2 (79.5% vs 20.5%). Synchronous delivery allowed students and lecturers to join sessions simultaneously at prearranged times, lectures were delivered in real time and interactions between lecturers and students were possible. Conversely, asynchronous delivery meant students accessed resources and listened to pre-recorded lectures at a time of their choosing. By T3, participants were given a third option allowing them to select a "mixed" approach with 54.8% preferring this option versus 41.2% for synchronous and 4.1% for an asynchronous mode of delivery.

Similarly, across all time points, more students expressed a preference for a blended approach to teaching that allows a mixture between university and remote delivery (57.6%, 56.8%, 62%) with less students selecting university (29.1%, 29.5%, 31.7%) or remote teaching (13.3%, 13.7%, 6.3%) alone. Concerning remote teaching and learning, free text responses indicated a preference for live and/or recorded lectures across all three time points. Benefits of live lectures included the ability to ask questions and receive feedback in real time and interactive opportunities. Technical glitches meant sometimes students missed out of class content. Recorded lectures were helpful as they could be listened to at a time suitable to the student to work at their own pace. This method meant they could pause, revisit material and make more detailed notes however not being able to ask questions was a common difficulty. Some students relayed feelings of finding it difficult to pay attention and keep motivated.

*"I like a mixture of recorded and live lectures, as it gives us a bit of variety and it is easier to stay engaged in the lecture. I like that with pre-recorded lectures, you can go at your own pace and pause it. . . ..live lectures make you feel more connected to the other students in your class and the lecturers. . . with the recording option, if you miss a live lecture you can still access it which is good". (Third Year, Speech and Language Therapy undergraduate student, Timepoint 1)*

Live sessions enabled other mechanisms for remote learning via the use of quizzes, worksheets, the chat function and discussion groups, some of which were also mentioned by the students.

The majority of students rated the overall quality of the platforms employed for remote teaching and learning for T1 and T2 as "excellent" or "average". Results from the most frequently used mediums can be seen in "Fig 1".

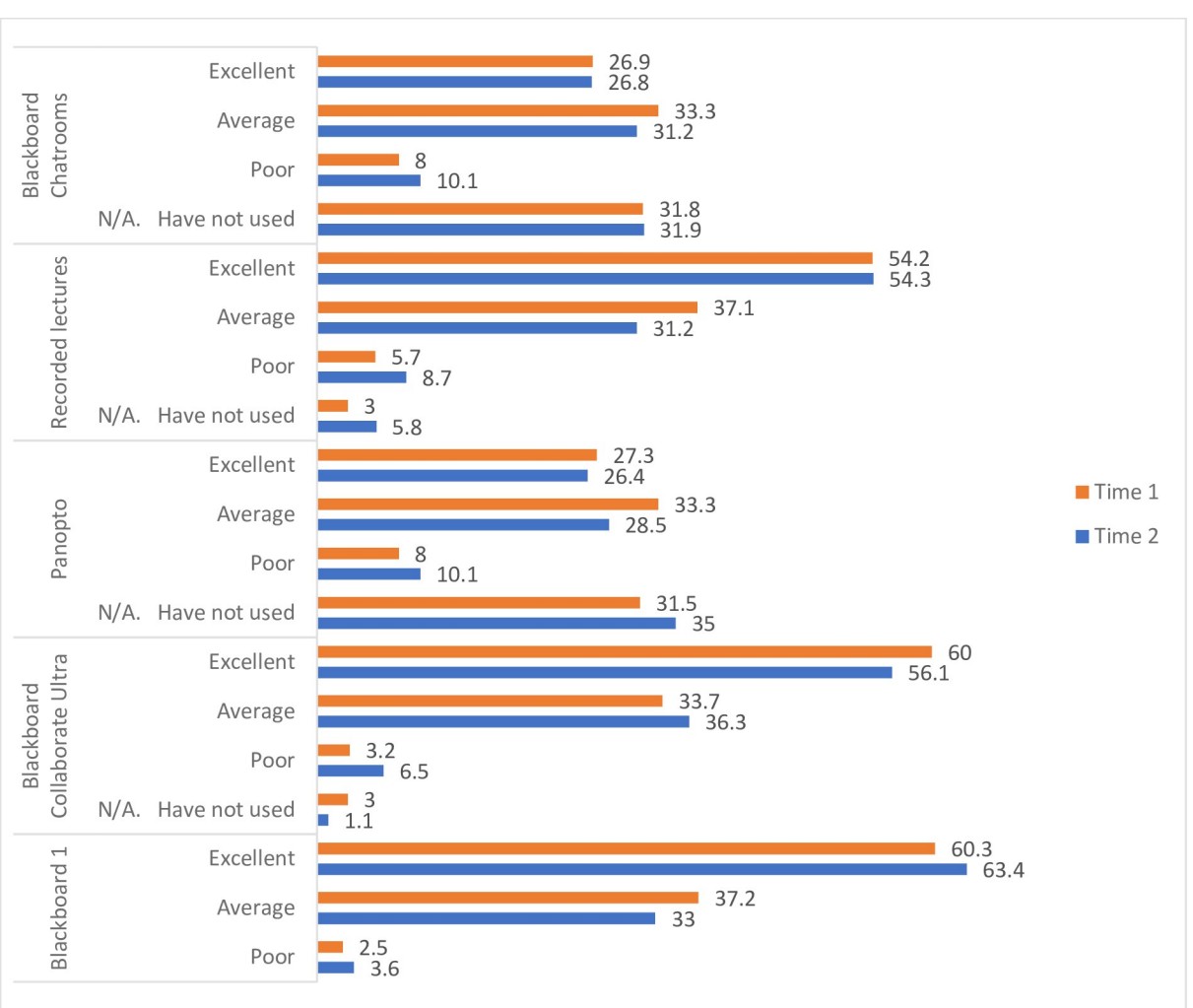

**Fig 1. Quality of platforms for teaching and learning selected by students (%) at Time 1 and Time 2.**

Other online platforms available but lesser used by students included Skype, Zoom, Nearpod, Microsoft Teams and Turning Point. Breakout rooms are sessions that are split off from main online platforms and allow participants to meet in smaller groups, where they are isolated in terms of audio and video from the main session. At T3, 94.1% of students had used these virtual rooms or performed groupwork online. Only 48.4% agreed that this method aided learning during remote delivery, with only 33% finding it easy to engage with their peers. Remote learning also impacted upon opportunities for peer learning during travel to university (41.3%) or during social interactions (77.5%). Whilst the majority of the sample were able to ask questions if they were unsure of any material (94.1%), 37.7% felt remote delivery impacted upon their ability to get a response from academic staff for course queries.

Students were asked about the impact of COVID-19 on their overall experiences at Time 1 and Time 2. Responses in relation to academic experiences are shown in "Fig 2". A higher percentage of students reporting negative effects of COVID across both time points with a minority of students (< 10.5%) reporting positive impacts. The largest negative impact of COVID was on the development of practical skills (78.8%, 70.8%) with some improvement over time, followed by the impact on the learning environment (62.2%, 61.1%). At T1 and T2, 43.3% and 44.1% of students respectively reported a negative impact on their assessments.

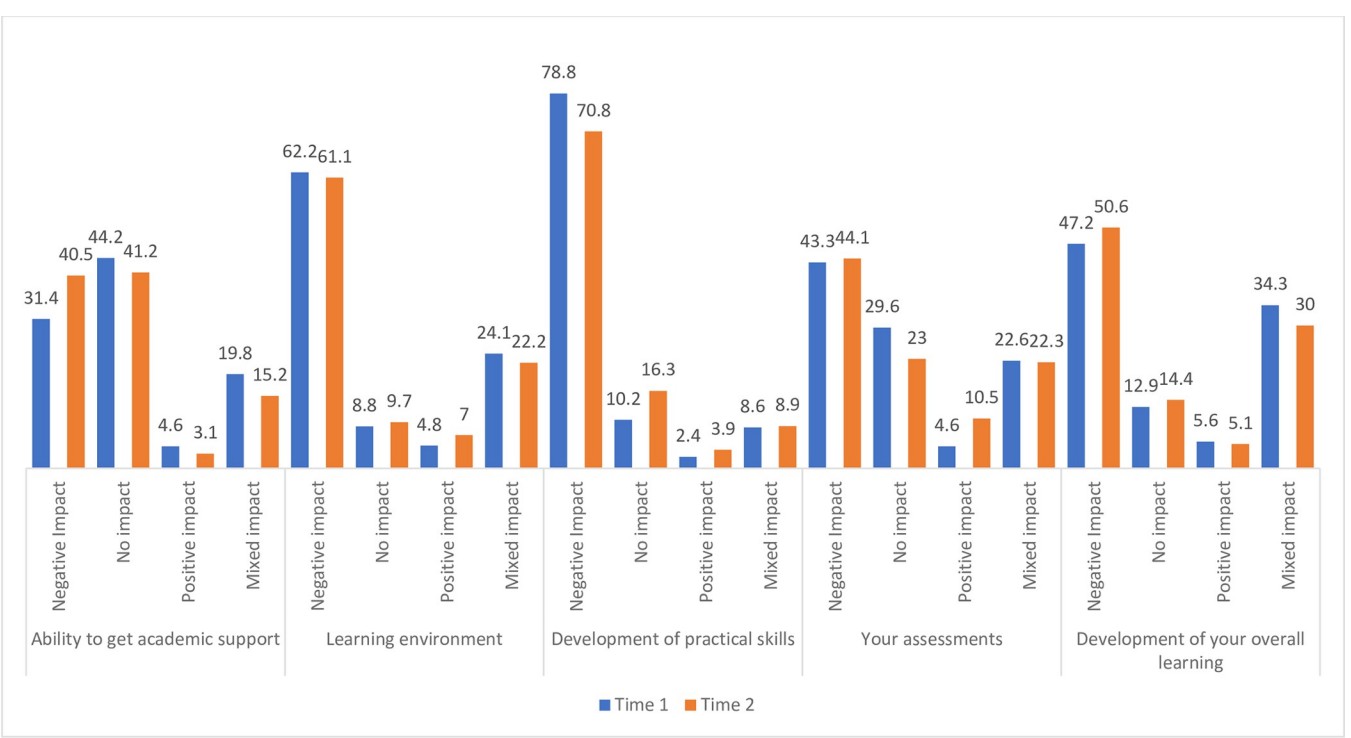

**Fig 2. Impact of COVID-19 on students' academic experiences (%) at Time 1 and Time 2.**

Students were also asked questions at T3 concerning the impact of online delivery on their assessments. For the majority of students (67.9%) assessments were delivered remotely. However, the findings suggest the assessment process was disrupted with only 23.5% informing their schedule was unchanged. Assessments were moved to different semesters (17.2%) and had been delayed or not yet completed (14.9%) for some of the respondents.

Students were also given the opportunity across all three timepoints to provide additional comments regarding the specific impact of COVID-19 on their learning and teaching. Similar to above, the pandemic had a mostly negative impact on the students that did not change over time. Practical challenges included WIFI access, screen time issues, getting a quiet place to work from home with no distractions and difficulties with concentration. Socially, they expressed concerns over a lack of engagement, difficulties making friends and missing informal interactions with other students for learning and peer support.

"*Socially I felt I was unable to fully gain the university experience and psychologically I was affected as I missed gaining many new friendships. . .*" (First Year, Diagnostic Radiography and Imaging, undergraduate student, Timepoint 3)

Academically, they experienced increased stress and difficulties managing their workload and reported challenges in maintaining self-discipline and motivation.

"*I feel extremely stressed at the moment. I'm finding it hard to separate study time from other responsibilities in the home. . .*" (Second Year, Speech and Language Therapy undergraduate student, Timepoint 1)

"*I feel like this year's workload is greater than last year, and sometimes it's been hard to manage..*" (Second Year, Speech and Language Therapy undergraduate student, Timepoint 1)

The development of practical skills via online learning was also a concern given the practical nature of their respective health professional roles.

*"I am just worried that I might not gain the practical skills required for my career..." (Second Year, Occupational Therapy undergraduate student, Timepoint 1)*

*"The lack of hands-on practical experience gives us an unfair disadvantage as it's much harder for us to grasp certain aspects of the course without more practical experience.." (First Year, Diagnostic Radiography and Oncology undergraduate student, Timepoint 2)*

Some students expressed how the pandemic had negatively impacted their mental health and their ability to learn. Overall, they worried about the impact on their grades.

Some students reported positive impacts in terms of developing new skills and gaining experience, not having to travel to the university and achieving a better work/life balance for those with family commitments.

*"As AHP students we can only benefit from working during a pandemic and we will have developed new skills for our professional career..." (Third Year, Radiotherapy and Oncology undergraduate student, Timepoint 3)*

## Clinical experiences

At the time of each survey, the minority of students (n = 64, n = 50 and n = 44) indicated they were on clinical placement. Information pertaining to the impact of the pandemic on clinical placement experiences "Fig 3" and skills "Fig 4" was collected across all three timepoints. In terms of clinical experiences, students reported an increasingly negative impact of COVID 19 over time concerning their integration into the unit/department (23.2%, 29.8%, 50%) and in particular the level of clinical supervision and feedback (12.7%, 29.8%, 66.7%). The negative impact of the pandemic on the opportunity to engage face to face with patients varied across time points with 49.2% selecting this response at T1, reducing to 34% at T2 and then increasing again to 46.5% at T3.

In terms of clinical placement skills, the pandemic also had an increasing negative impact on the development of professional (23.2%, 27.7%, 51.2%) and interpersonal skills (22.8%, 23.9%, 64.3%) as well as the assessment of skills and knowledge (19%, 31.9%, 66.7%). The

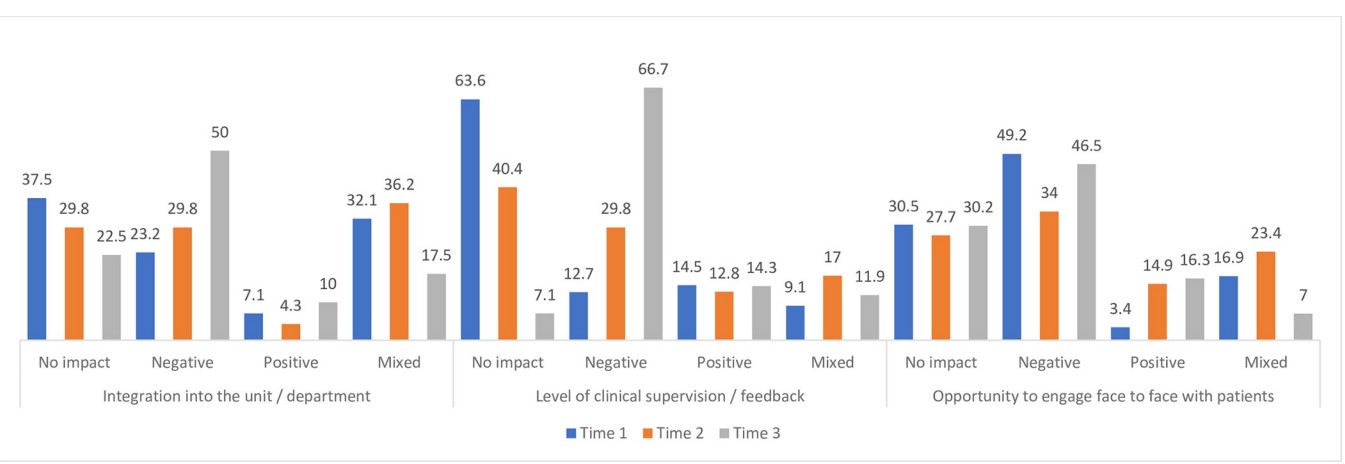

**Fig 3. Impact of COVID-19 on students' clinical placement experiences (%) at Time 1, Time 2 and Time 3.**

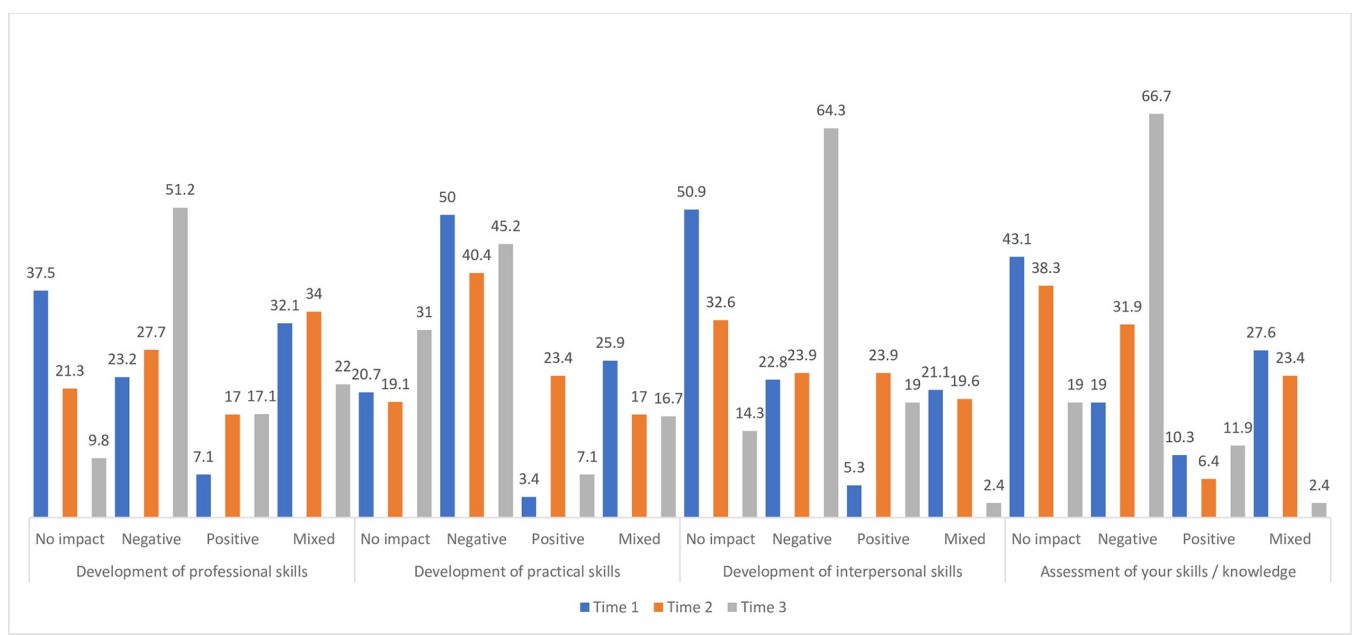

**Fig 4. Impact of COVID-19 on students' clinical placement skills (%) at Time 1, Time 2 and Time 3.**

majority of students reported a negative impact on the development of practical skills across all three timepoints with 50% selecting this response at T1, then reducing to 40.4% at T2 before increasing again at T3 to 45.2%.

At T3, 51.2% felt limited in the places/departments/rooms they could attend due to social distancing measures with only 37.2% feeling they were not restricted and could see everything their colleagues were doing. Only 32.6% felt the university fully prepared them for placement. Nevertheless, 44.2% felt their confidence had increased due to seeing challenging cases. Just over half (55.8%) of students found placement during a pandemic a rewarding experience.

Free text responses indicated the main impact of COVID 19 on placement reported by students came as a direct or indirect result of infection control. Staff shortages due to self-isolation meant less time for teaching, increased cleaning for remaining staff reduced the time they had to explain things to students. There was also a reduced capacity for students to see patients. Some departments had less variety in the types of patients' students were seeing, some areas did not want students, had no room due to social distancing or some had to shut down completely. This in turn impacted on learning skills and the development of practical skills including being able to use or understand how to work certain pieces of equipment.

> "Covid has impacted so much on my placement that I fear I may have to seek advice to take a gap year. . . .I was in a smaller department that could not hold two students due to social distancing therefore I only attended placement two days per week. . .I lost another week of placement due to my department shutting down due to COVID. . ." (Second Year, Health Science Physiology undergraduate student, Timepoint 1)

> "The biggest impact is not being able to use certain pieces of equipment. . ..also staff shortages due to self-isolation has left less time for teaching.." (Third Year, Physiotherapy undergraduate student, Timepoint 2)

> "Very limited face to face contact with patients. Unable to develop practical treatment skills. (I) feel like I would be learning a lot more if COVID didn't exist. It makes me worried for

*when I graduate, I won't be fully prepared..*" (Third Year, Physiotherapy undergraduate student, Timepoint 2)

Some students informed missed placements have been rescheduled for a later date, though this may conflict with their academic studies thus increasing the burden of their workload.

Aside from learning, the pandemic impacted upon students' wellbeing on placement in terms of increased stress and worry that they may catch COVID-19 or pass this on to others.

"*I have limited social interactions and I have a constant worry of contracting and passing on COVID from the healthcare settings I am currently working in for placement. . ..*" (Second Year, Speech and Language Therapy undergraduate student, Timepoint 3)

Those on placement felt reassured by PPE, vaccinations, being of personal low risk and some felt less concerned about COVID over time. Regarding PPE there was a general improvement over time concerning students' feelings that they had: adequate access to PPE (66.2%, 78.4%, 97.6%), sufficient knowledge about appropriate PPE (57.6%, 78.4%, 95.2%) and sufficient training on use, donning and doffing (47.2%, 68.5%, 92.9%). Over T2 and T3 students reported less problems concerning the fit testing of masks for T2 and T3 (18.1%, 14.3%). However, across all three timepoints, there was an increase in problems wearing PPE in terms of allergies/irritations (14.1%, 20%, 26.2%). All students responded to this information for T1 and T2. Only those on clinical placement responded at T3.

## Personal experiences, attitudes and opinions

A minority of students tested positive for COVID-19, which increased slightly across the three time points "Fig 5". The majority of students confirmed they did not have to isolate at T2 and T3, with very little change between the two time points (65.8% versus 62.6%).

The majority of respondents knew they had to isolate having either had symptoms (6%, 7.9%), had symptoms and a positive test (17%, 25.8%), been notified through friends, family or

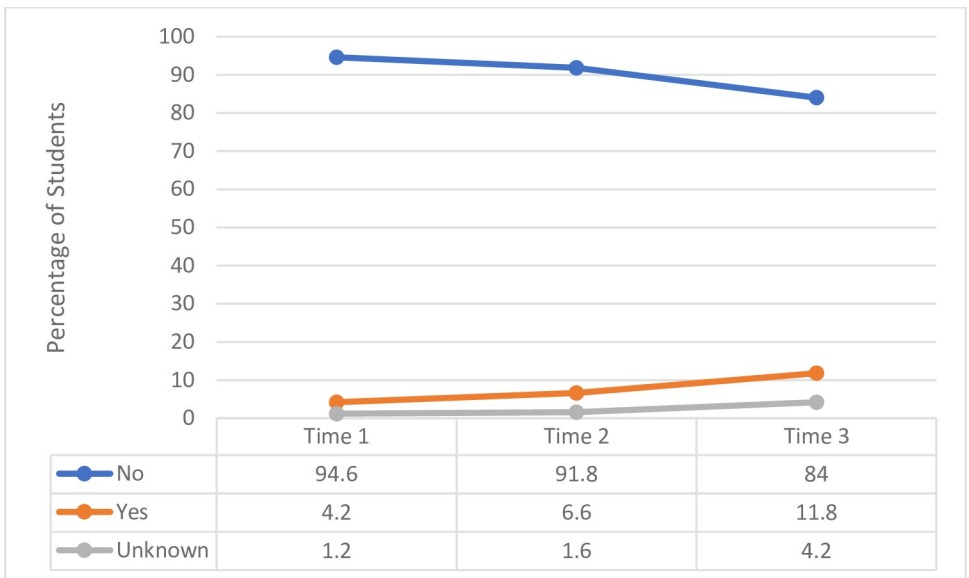

| | Time 1 | Time 2 | Time 3 |
|---|---|---|---|
| No | 94.6 | 91.8 | 84 |
| Yes | 4.2 | 6.6 | 11.8 |
| Unknown | 1.2 | 1.6 | 4.2 |

**Fig 5. COVID-19 test results for the sample of students (%) throughout the study for Time 1, Time 2 and Time 3.**

work colleagues (66%, 70.8%) or they were notified through the phone app (11%, 23.6%). For those who indicated they did have to isolate, this impacted upon learning and teaching in terms of missing university (23%, 13.6%), missing clinical placement (12%, 26.1%) and falling behind with their studies (27%, 23.9%). At T3, 67.3% of students reported they had already received the vaccine "Fig 6".

General thoughts and feelings about COVID-19 were examined across T1 and T2 "Fig 7" as well as opinions having lived with the pandemic for one year at T3 "Fig 8". The main concern reported by students was the worry that people they loved or cared about would become sick. This finding remained stable across T1 (77.1%) and T2 (77.8%). Almost half reported finding it easy to go out as long as they adhered to social distancing, although 32.4% reported trying to avoid contact as they personally did not want to become ill at T1, increasing to 43.7% at T2. At T3 the majority of students (73.3%) reported feeling fed up living a socially distanced lifestyle and a desire to return to normal. The main concern at this time point related to the amount of time sitting at a computer and staring at a monitor (61.5%) with only 22.6% of students reporting being able to control their screen time and manage regular breaks throughout the day.

During T1 and T2 participants were asked a range of questions exploring the impact of COVID-19, some of which extended to their personal experiences "Fig 9". The pandemic had no impact on accessing technology for the majority of respondents (64.8%, 67.7%) and a more varied response of mainly no impact or a negative impact on personal and financial circumstances. The majority of students reported negative impacts of COVID in terms of making and maintaining friendships (63.6%, 69.3%) and mental wellbeing (55.5%, 59.1%) that increased slightly over time.

In another section of the survey the majority of students reported that they were using strategies to keep themselves physically and mentally well, a finding that remained stable across all time points (72.8%, 69.3%, 70.2%). Students commented on using a wide range of activities to keep physically and mentally fit. Free text comments indicated students were often using more than one strategy. Physical strategies including various types of exercise and sporting activities with some students referring to the benefit of physical activity for their mental health. Psychological strategies including mindfulness, meditation, relaxation and breathing techniques.

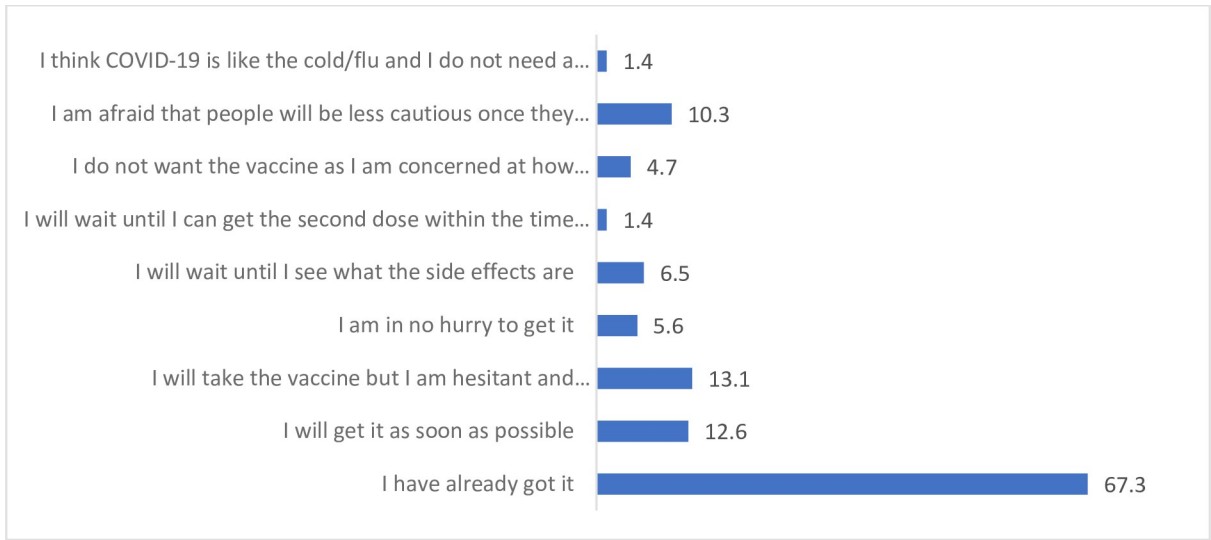

**Fig 6. Students (%) thoughts and feelings about the vaccine at Time 3 (n = 214)***. * Percentages do not add to 100% as participants could select multiple responses.

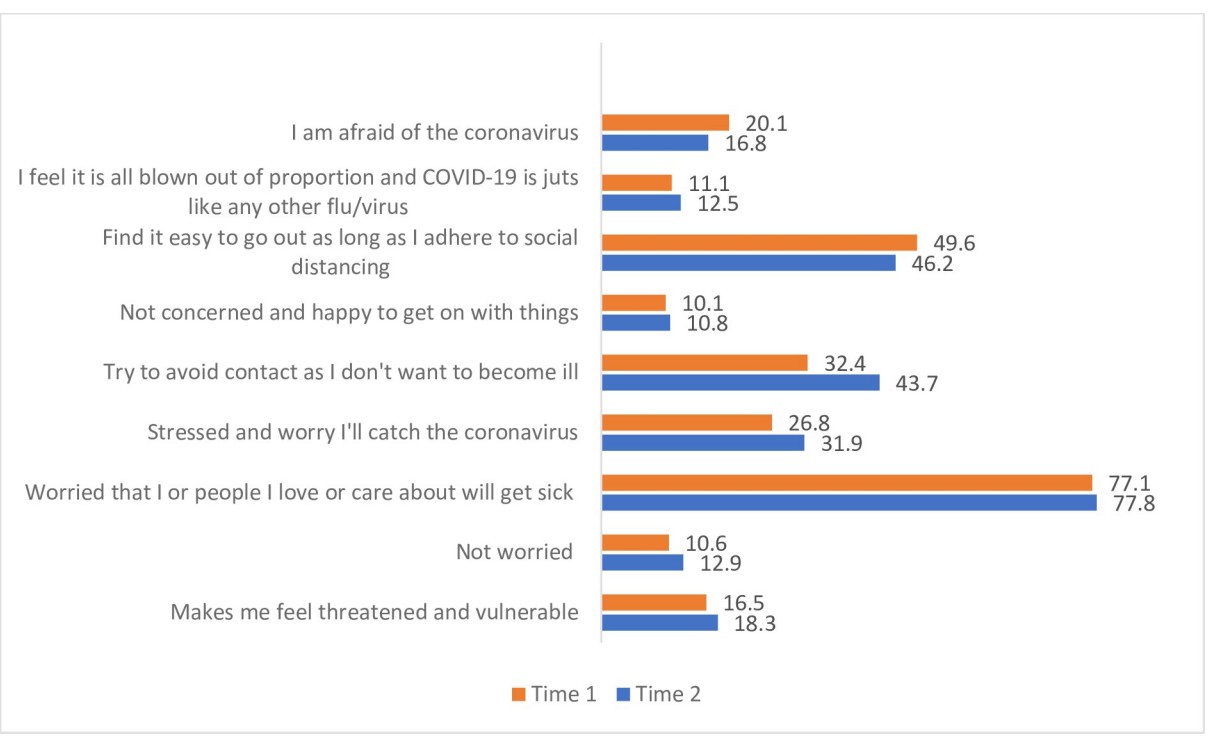

**Fig 7. Students (%) general thoughts and feelings about COVID-19 at T1 (n = 407) and T2 (n = 279)**[*]. Percentages do not add to 100% as participants could select multiple responses.

Students also mentioned participating in various hobbies, connecting with friends and family face to face and online, monitoring their daily routines to ensure healthy eating and sleep patterns, limiting screen time, news and social media and making time for oneself. Some students informed they had sought professional help accessing medication, counselling and therapy.

The majority of students reported that the coverage of the pandemic on news and/or social media had no impact on their thoughts about a career as a healthcare professional or their thoughts about clinical placement across all three time points. However, although a higher percentage of respondents indicated a positive effect as opposed to a negative effect on their career choice, the opposite was the case for the effect on their thoughts about clinical placements that decreased over time "Fig 10".

Students were also asked to provide further details on how COVID-19 overall had influenced their choice of profession. For the majority of students COVID 19 either had no impact or a positive impact on their choice of career. It gave them a real-world view of their profession and reinforced their choice. Some commented on how the health service has been portrayed in a positive light with an increased respect and gratitude for healthcare workers.

> *"I still want to pursue a career as a healthcare professional and if anything, the pandemic has made me want to pursue it more so due to the now even higher level of respect that I have for healthcare professionals looking after people during the pandemic and putting themselves and their families at risk.." (Second Year, Diagnostic Radiography and Imaging undergraduate student, Timepoint 1).*

> *"I feel it has shown the value of the healthcare profession and it has highlighted the vital work that we do. . ." (Second Year, Speech and Language Therapy undergraduate student, Timepoint 3)*

Some students felt more valued in their career choice and being part of the pandemic effort, determined to help.

*"COVID-19 is affecting all aspects of our lives, but we must take any chances to progress in our learning and on placement. We are involved in healthcare and healthcare must go on…"* (First Year, Radiotherapy and Oncology undergraduate student, Timepoint 3)

Some students acknowledged the positive impact of COVID 19 on their ability to adapt, whilst others voiced pressures to learn quickly in unforeseen circumstances. For a few, the pandemic had a negative impact on their career choice. Others commented on feelings of not being appreciated and taking risks working for zero reward financially or academically.

*"Students have to risk getting covid and working in the hospital and not even getting paid.."* (Third Year, Radiotherapy and Oncology undergraduate student, Timepoint 2)

*"(I'm) now more concerned about taking the risk for zero financial reward….(Third Year, Diagnostic Radiography and Imaging undergraduate student, Timepoint 2).*

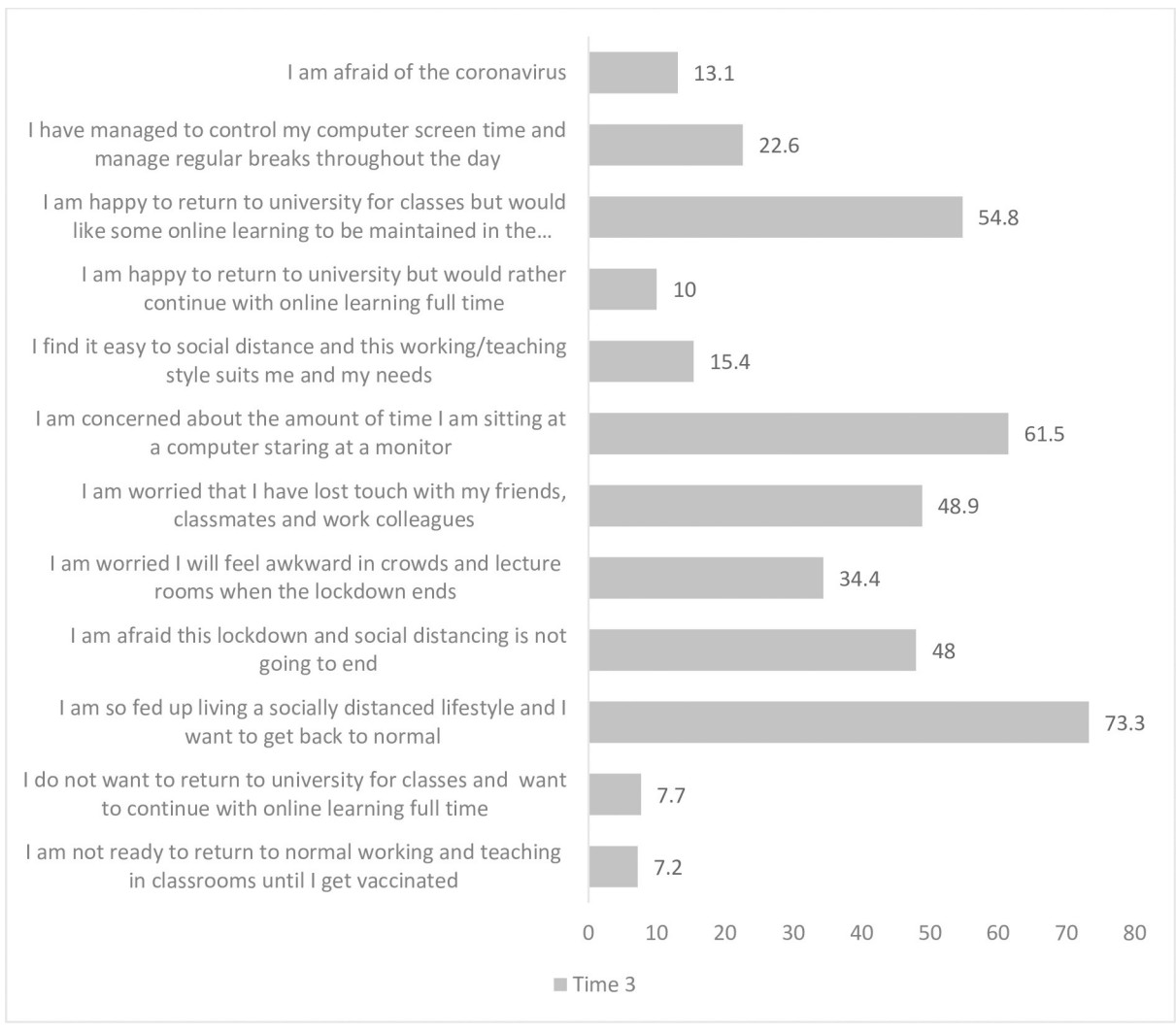

**Fig 8. Students (%) opinions after living with COVID-19 for one year at T3 (n = 221)\***. Percentages do not add to 100% as participants could select multiple responses.

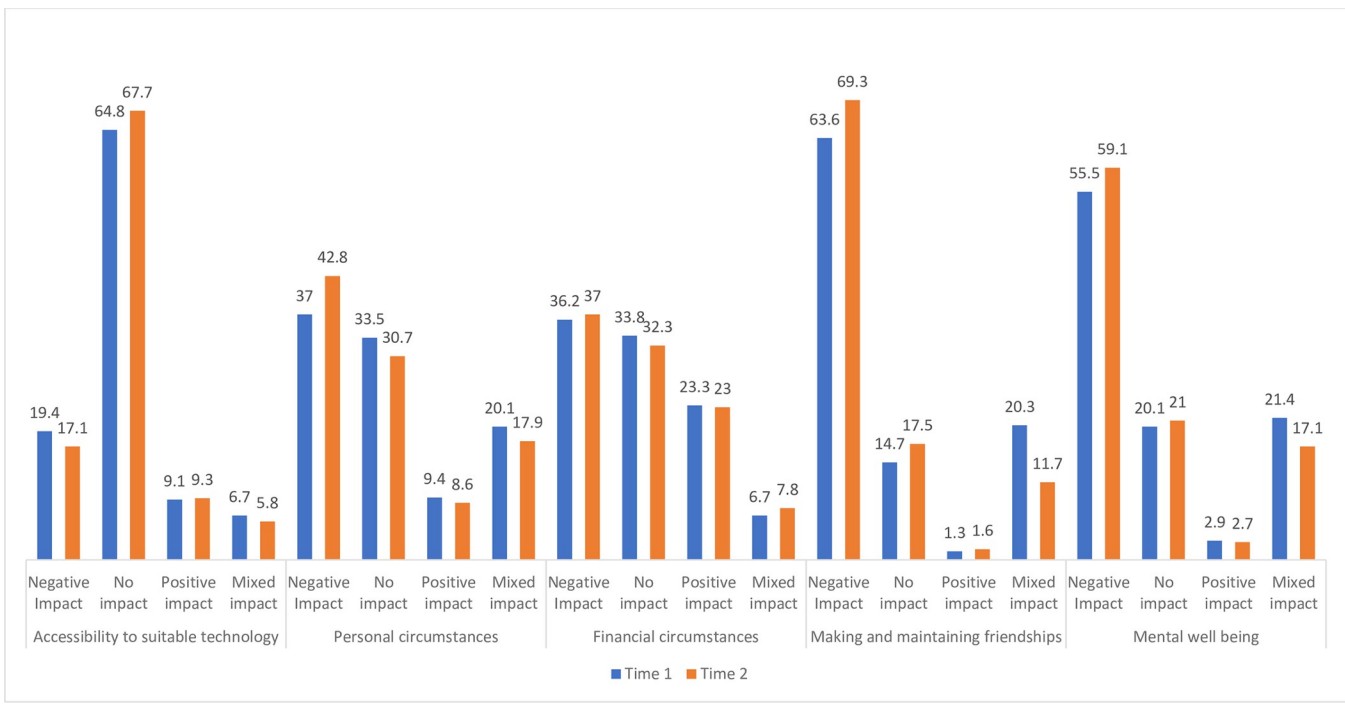

**Fig 9. Impact of COVID on personal experiences.**

## Discussion

The aim of this study was to explore the impact of COVID-19 on the academic, clinical and personal lived experiences of healthcare students. The core business of any university is the delivery of student education and thus, learning and teaching, how it is delivered through the

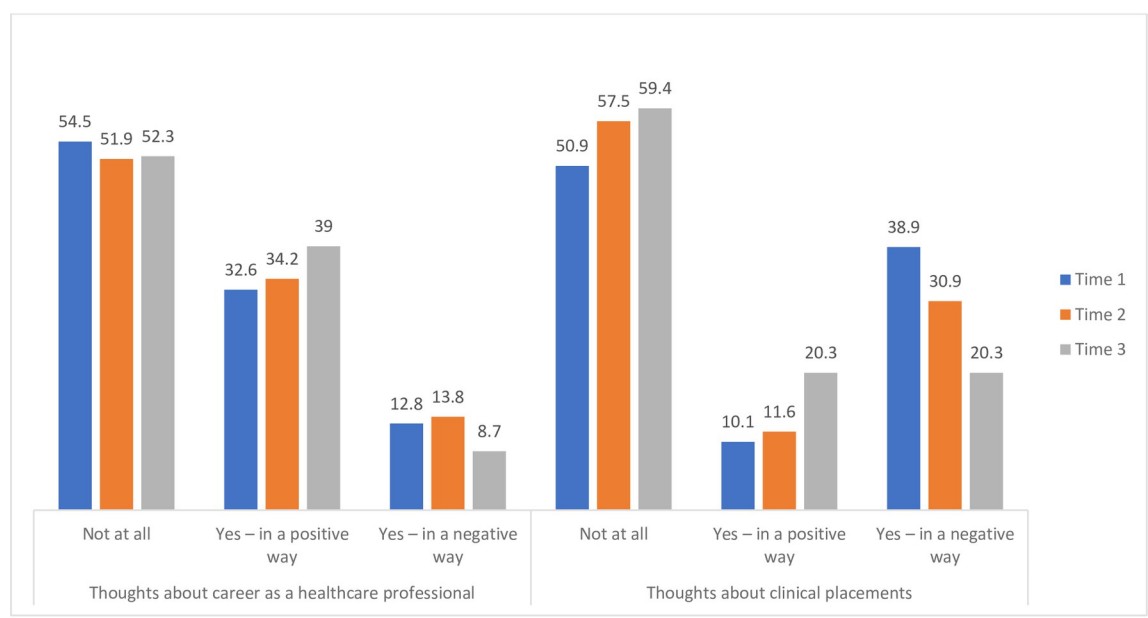

**Fig 10. Impact of news and/or social media coverage of COVID-19 on career choice as a healthcare professional and thoughts about clinical placement (% students).**

use of TELs and VLEs, and understanding barriers to student engagement during a global pandemic is fundamental to successful outcomes for students. Timely and successful completion of courses becomes more important at a time when pressure on the Health and Social Care system and its staff require a continued throughput of students to sustain the workforce.

At the time of each respective survey, a greater number of students were in university rather than on clinical placement. Although the majority of students had access to a laptop suitable for academic work and mostly rated platforms for remote learning as excellent or average, challenges in relation to having access to reliable fast broadband, a desk suitable for academic work and a quiet/adequately sized working space impacted upon the experience of remote learning for many students. During remote learning, the benefits of live lecturers included the ability to ask questions, receive a response in real time and increased opportunities for interaction. Recorded lectures were more beneficial for revision purposes allowing students to work at their own pace, pause, make detailed notes and revisit material at a later date, however students reported difficulties maintaining attention and motivation. At Time 1 and Time 2, a higher percentage of students reported negative impacts of the pandemic on the learning environment, the ability to get support, and the development of their overall learning. The largest negative impact of the pandemic was on the development of student practical skills in the university setting. Although the focus of university learning centres on the acquisition of theory and knowledge, the development of practical skills is a vital aspect of learning given the nature of their respective healthcare roles. Previous research using OSCE (objective-structured clinical examination) reported the pandemic hampered learning outcomes among final year medical students in their clinical preparation [18]. Other research has suggested an increased need to counterbalance the lack of clinical experience with virtual or simulated learning [19].

The assessment process was delayed or disrupted for the majority of students, which may have implications for increasing the burden of their workload at later stages of their studies. These findings were echoed in free text responses gathered across all three survey time points. Students reported challenges working from home due to technical and situational difficulties with increased feelings of stress and difficulties managing their workload. They missed informal social interactions with their peers as opportunities for learning and support. Despite the majority of students using breakout rooms, only a third found it easy to engage with their peers in this setting. Given the impact of the pandemic on peer relations and the ability to access staff, it is vital for universities to find ways of improving interactions in remote learning environments, particularly when such interactions are important for enhancing mental health and wellbeing [15].

The results highlighted a predominance of the use of one digital platform and the tools within it. The limited use of a variety of platforms can be attributed to several factors. At the onset of the pandemic, institutional guidance and training focused on the use of the University's VLE, Blackboard Learn and its functions, such as Blackboard Collaborate Ultra and Panopto. Privacy and General Data Protection Regulation (GDPR) concerns prevented or discouraged the use of other digital platforms or applications and in addition to lecturer unfamiliarity resulted in limited adoption. This is evident in the results where between 63.41% and 90.27% of respondents had not used the platforms or applications which sat outside the Blackboard Learn suite. Lower use of Blackboard chat rooms (Approximately 32% of participants had not used chat rooms at Time 1 and Time 2) may have been a missed opportunity to maximise engagement with Generation Z students, viewed as 'digital natives' [20] who are familiar with the use of digital participatory spaces [21]. Nevertheless, going forward they expressed a preference for a more flexible approach to their academic experiences using a blended approach of synchronous and asynchronous course delivery and remote and university based learning challenging universities, academics and students to maximise engagement with digital

technologies. Academics should be encouraged and supported to widen their use of digital technologies with ongoing training in technologies which meet the needs of students within specific AHP and HCS programmes. This requires a co-produced strategy to adequately capture and implement staff and student technology requirements.

As previously mentioned, less students were on clinical placement at the time of each survey. Previous research has reported negative impacts of the pandemic on student's medical training [19] and confidence in their clinical skills [22]. In the current study, the pandemic had an increasingly negative impact on their clinical skills and experiences in terms of their integration into the department, the opportunity to engage with patients, their professional and interpersonal skills and on the assessment of their skills and knowledge. Although this impact increased over all three time points, the difference between each interval was greatest between times 2 and 3 ranging from between 20–40%. A negative impact was also reported for the development of practical skills and opportunities for face-to-face patient engagement, which showed similar trends in responses over time with the highest response rates at Time 1, reducing at Time 2 and then increasing again at Time 3. Free text responses indicated lost placement time, less time for teaching and reduced opportunities for varied face to face patient contact, all of which impacted on the development of practical skills. At Time 3, approximately half (51.2%) felt limited in the places/departments/rooms they could attend and only 37.2% felt they could see everything their colleagues were doing. Observational learning is also an important aspect of the student learning experience. Given the negative impact of COVID-19 on the development of practical skills during university learning and clinical placement, consideration should be given to how this could be improved.

During clinical placement, students reported increased stress and worry that they may catch or transmit COVID-19. A minority of students tested positive for COVID-19 throughout the surveys (<12%) and just over a third had to self-isolate at Time 2 and Time 3. At Time 3 just over two thirds of students confirmed they had received the vaccine. Similar to previous research involving healthcare staff [23–25] students felt reassured by PPE, with improvements across time in terms of access, knowledge and training. There was however increased rates of allergies and irritations associated with its use.

Beyond clinical placement, the concern that people they loved or cared about would become sick remained a stable finding across time 1 and 2 for the majority of students, a finding similar to previous research [17]. The impact of lockdown and online learning has resulted in a substantial proportion of students reporting a negative impact on their friendships, affecting their ability to form and maintain relationships. The participants were in first, second and third year of healthcare courses and each stage brings with it different challenges in the formation, development and maintenance of friendships. Boda et al, 2020 [26] discuss the importance of physical proximity and random encounters in the development of friendships which are missing during lockdown and restricted interactions. Globally, higher education institutions have concerns regarding the potential negative impact of the COVID19 pandemic upon students' mental wellbeing. A recent UK study reported a decrease in self-reported student mental wellbeing during the first 5 weeks of "lockdown" [27]. Similarly, increased levels of anxiety, depression and distress were noted amongst French students during the initial months of the pandemic [28].

It is also notable that 73% of respondents were using strategies to keep physically and mentally well, which may have affected the reported impact of the pandemic on their mental wellbeing. Coping strategies for physical and mental well-being were mainly exercise and keeping in contact with friends and family; these as well as additional strategies that have been reported such as mindfulness, positive self-talk and spiritual support [29] could be shared with other students so that they too could consider whether they would be useful to help them cope.

Coping strategies could be embedded in the curriculum so that students are better equipped to manage their well-being during high or unexpected periods of stress. Whilst it was not always clear if students had always participated in these strategies prior to the pandemic, some commented on the impact of the pandemic on their mental health despite taking action to keep well. Some students reported seeking professional help, no one referred to accessing university student support. Timely mental health care and mental healthcare training need to be developed and implemented as part of professional development activities [30].

In a study of public perceptions of social distancing and isolation during the covid-19 pandemic, participants reported losses of social interaction, income and structure and routine leading to psychological impacts, loss of motivation, loss of meaning and loss of self-worth [31]. In the present study, students echoed feelings of isolation and loss of interaction, routine and motivation. It is important to consider the impact of the pandemic on the personal lives of students beyond university and clinical settings that may in turn impact their learning. The pandemic had a more varied response of mainly no impact or a negative impact on financial circumstances. The financial burden acquired during the course of university degree can under normal circumstances negatively impact upon students, and many students are reliant upon part time employment to supplement their income. A recent Canadian study noted that redundancies and loss of job opportunities caused by the pandemic has strongly impacted students, with many having concerns about debt accrual [32]. In the present study, 87% of students responded that they have part time employment. At Time 1, 61% of students reported a negative impact of the pandemic on their finances during clinical placement, which reduced over time. Beyond placement, some students reported positive impacts of the pandemic including the furlough scheme, being able to increase working hours and reduced travel and living costs.

Reassuringly, for the majority of students COVID 19 either had no impact or a positive impact on their choice of career. For others however, the pandemic had a negative impact on their career choice. Some commented on feelings of not being appreciated and taking risks working for zero reward financially or academically. Previous studies with healthcare staff during the MERS-CoV identified staff appreciated employer recognition for their efforts, which would entice them to work in future epidemics [25]. As a contributing part of the future healthcare workforce, it is essential students feel valued in their roles.

## Strengths and limitations

The authors highlight caution that the findings may not be generalisable beyond the study sample in Northern Ireland. The study sample consisted of healthcare students only, who across all three timepoints of the survey were more likely to be female, aged less than 23 years old and less likely to have caring responsibilities or live with someone who was shielding or high risk, which may also limit the generalisability of the findings. Another limitation of the results was due to smaller sample sizes for time 2 and time 3 and the inability to conduct a longitudinal analysis as unique identifying information was not routinely collected. The study also did not utilise objective measures of academic performance, clinical skills or psychological well-being. Results were based only on descriptive statistics. Study strengths include the timing of the surveys to provide a snapshot of student experiences at multiple time points throughout the pandemic during the prevalence of increasing infection rates and when increased lockdown restrictions were in place. The data collected from this study also provides research into the specific experiences of students during a pandemic as opposed to staff, which has been limited in the literature.

## Conclusion

Given the potential for future and continued pandemic restrictions, as well as an expressed preference for a blended approach to academic learning going forward, universities should consider how learning programmes can meet the needs of students in terms of the development of their practical skills, the ability to access support, manage their screen time and workload as well as being able to maintain engagement and provide opportunities for peer interactions. As the development of practical skills was also a key concern in the clinical setting, consideration should be given to increase the status and movement of students on placement akin to healthcare staff to ensure maximum benefit is gained from the placement setting. The current study provides data from a range of different student healthcare disciplines at all three stages of learning, collected at multiple timepoints during a full academic year amidst the pandemic. These findings can be considered by educators to ensure students have the adequate skills and knowledge to perform their professional roles. Finally, with the potential interconnecting impact of academic, clinical and personal experiences, educators should adopt a more holistic approach to student wellbeing, ensuring students are aware of internal and external support mechanisms.

## Supporting information

**S1 File. Copy of Qualtrics COVID undergraduate student survey at timepoint 1.**
(PDF)

**S2 File. Copy of Qualtrics COVID undergraduate student survey at timepoint 2.**
(PDF)

**S3 File. Copy of Qualtrics COVID undergraduate student survey at timepoint 3.**
(PDF)

## Author Contributions

**Conceptualization:** Sonyia McFadden, Jean Daly-Lynn, Brenda O'Neill, Joanne Marley, Catherine Hanratty, Paul Shepherd, Lucia Ramsey, Cathal Breen, Orla Duffy, Andrea Jones, Daniel Kerr, Ciara Hughes.

**Data curation:** Sonyia McFadden, Sharon Guille, Jean Daly-Lynn, Ciara Hughes.

**Formal analysis:** Sonyia McFadden, Sharon Guille, Jean Daly-Lynn, Ciara Hughes.

**Investigation:** Sonyia McFadden.

**Methodology:** Sonyia McFadden, Jean Daly-Lynn, Brenda O'Neill, Joanne Marley, Catherine Hanratty, Paul Shepherd, Lucia Ramsey, Ciara Hughes.

**Project administration:** Sonyia McFadden, Jean Daly-Lynn, Ciara Hughes.

**Resources:** Sonyia McFadden.

**Software:** Sonyia McFadden, Sharon Guille.

**Supervision:** Sonyia McFadden.

**Validation:** Sonyia McFadden, Jean Daly-Lynn, Ciara Hughes.

**Visualization:** Sonyia McFadden, Jean Daly-Lynn, Ciara Hughes.

Writing – original draft: Sonyia McFadden, Sharon Guille, Jean Daly-Lynn, Brenda O'Neill, Joanne Marley, Catherine Hanratty, Paul Shepherd, Lucia Ramsey, Orla Duffy, Andrea Jones, Daniel Kerr, Ciara Hughes.

Writing – review & editing: Sonyia McFadden, Sharon Guille, Jean Daly-Lynn, Brenda O'Neill, Ciara Hughes.

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
