## [Decision Letter · Decision Letter 0]

18 Jan 2022

PONE-D-21-29579The impact of the COVID-19 pandemic on healthcare students.PLOS ONE

Dear Dr. McFadden,

Thank you for submitting your manuscript to PLOS ONE. After careful consideration, we feel that it has merit but does not fully meet PLOS ONE’s publication criteria as it currently stands. Therefore, we invite you to submit a revised version of the manuscript that addresses the points raised during the review process.

We look forward to receiving your revised manuscript.

Kind regards,

Kamran Sattar

Academic Editor

PLOS ONE

Journal Requirements:

2. Please consider changing the title so as to meet our title format requirement (https://journals.plos.org/plosone/s/submission-guidelines). In particular, the title should be "Specific, descriptive, concise, and comprehensible to readers outside the field" and in this case it is not informative and specific about your study's scope and methodology.

6. Please amend the manuscript submission data (via Edit Submission) to include author Sharon Guille, Jean Daly-Lynn, Brenda O’Neill, Joanne Marley, Catherine Hanratty,  Paul Shepherd, Lucia Ramsey, Cathal Breen, Orla Duffy, Andrea Jones, Danny Kerr, Ciara Hughes.

7. Please amend your list of authors on the manuscript to ensure that each author is linked to an affiliation. Authors’ affiliations should reflect the institution where the work was done (if authors moved subsequently, you can also list the new affiliation stating “current affiliation:….” as necessary).

Additional Editor Comments:

Dear Authors, Thank you submitting your manuscript. Reviewers have gone through your work, and made suggestions/comments. You are now requested to read all the comments (concerning; Abstract, Introduction, Subjects and Methods, Research design, Setting and participants, Results, Discussion, Limitations). These all sections need a good revision. Moreover, to avoid confusion for the readers, I suggest authors rearrange all the tables and the qualitative results should be addressed in light of the participants' or groups' responses. Also adding details as, how will your findings contribute to the present body of knowledge? shall make your current work robust.

Reviewers' comments:

Reviewer's Responses to Questions

**Comments to the Author**

1. Is the manuscript technically sound, and do the data support the conclusions?

Reviewer #1: Yes

Reviewer #2: Yes

2. Has the statistical analysis been performed appropriately and rigorously? 

Reviewer #1: Yes

Reviewer #2: No

3. Have the authors made all data underlying the findings in their manuscript fully available?

Reviewer #1: Yes

Reviewer #2: No

4. Is the manuscript presented in an intelligible fashion and written in standard English?

Reviewer #1: Yes

Reviewer #2: Yes

5. Review Comments to the Author

Reviewer #1: Authors have explained the idea very well. They used descriptive statistics to describe the impact of COVID-19 on their academic, clinical and personal experiences. Please remove the objectives , written twice. Need some minor grammar corrections / reference. They could use also chi-square / ANOVA analysis to see the relationships among groups.

Reviewer #2: Dear Authors,

Thank you for the opportunity accorded to me in reviewing your manuscript. I understand the merit of your study. Please feel free to refute or agree with my comments/suggestions.

Keep safe!

Abstract:

Please remove abbreviations. If you will use abbreviations, please spell it out first (i.e, COVID-19). This will make your abstract clear.

The abstract needs to be presented clearly and succinctly.

Introduction:

Limit the discussion on COVID-19.

Provide more literature on the effect of online learning to the psychological well-being of healthcare students. This is the context of your study.

I do not see the dearth in the manuscript that will establish the reason why this kind of study needs to be conducted. There is a plethora of studies like your study. Why there is a need to replicate it again.

What is the contribution of this study to medical science. Please state.

Subjects and Methods:

Research design: Rewrite to make it understandable.

Setting and participants:

Please state clearly the study setting. Describe it based on its characteristics that directly relates to your study (example: when it was founded, location, how many students). How did your select your participants? Inclusion and inclusion criteria? Sampling technique? Attrition rate? Completed the survey? The section needs a good revision.

Setting: This heading is a discussion of the instrument and data collection and not the setting.

Statistical analysis: What is the assumption why this statistical analysis was used to treat the data considering the variables being studies?

Results: There needs to be a clear presentation of this section. What was the number of students in the 1st, 2nd, and 3rd years, as well as the response rate that must be reported in the manuscript? In table 1, I did not find the "vast majority 80.6% of females." and many more. I suggest authors rearrange all the tables. If anyone wants to cross-reference the data in the table and results section, it is readily available.

There is more confusion in the results section because of the mixed methods used. Explain that qualitative results should be addressed in light of the participants' or groups' responses. The authors did not elaborate on the statement with which participants reported this finding. Another suggestion for authors is to conduct more analysis to reach a conclusion.

Discussion: This section does not need to entirely state the results. Please create an argument of your results to the previous literature. Identify controversies, support your findings, negate your findings based on the existing literature.

How will your findings contribute to the present body of knowledge? This should be evident throughout your discussion. Readers need an analytical presentation of this section.

Limitations: Please add more limitations as your study had many.

6. PLOS authors have the option to publish the peer review history of their article (what does this mean?). If published, this will include your full peer review and any attached files.

Reviewer #1: No

Reviewer #2: No

---

## [Author Response · Author response to Decision Letter 0]

31 Mar 2022

The following have been uploaded as separate files.

Rebuttal letter that responds to each point raised by the academic editor and reviewer(s). You should upload this letter as a separate file labeled 'Response to Reviewers'.

---

## [Decision Letter · Decision Letter 1]

12 Apr 2022

PONE-D-21-29579R1The impact of the COVID-19 pandemic on healthcare students.PLOS ONE

Dear Dr. McFadden,

Thank you for submitting your manuscript to PLOS ONE. After careful consideration, we feel that it has merit but does not fully meet PLOS ONE’s publication criteria as it currently stands. Therefore, we invite you to submit a revised version of the manuscript that addresses the points raised during the review process.

We look forward to receiving your revised manuscript.

Kind regards,

Kamran Sattar

Academic Editor

PLOS ONE

Additional Editor Comments:

Dear Authors,

Thank you for the revised file of your manuscript and making the required changes based on the comments.

Following are few more suggestions about how to improve the reporting of observational studies and facilitate critical appraisal and interpretation of studies by reviewers, journal editors, and readers.

A lot of medical research is based on observation and to avoid observational studies' reporting to be compromised for the quality, we recommend using STROBE checklist.

Please ensure the reporting of your observational study using the STROBE checklist (http://www.strobe-statement.org). The STROBE Statement consists of a checklist of 22 items, which relate to the title, abstract, introduction, methods, results, and discussion sections of articles.

Under Discussion, abbreviated terms 'TLEs and VLEs' are used without being spelled in full elsewhere. Please correct.

Best Regards!

Reviewers' comments:

Reviewer's Responses to Questions

**Comments to the Author**

1. If the authors have adequately addressed your comments raised in a previous round of review and you feel that this manuscript is now acceptable for publication, you may indicate that here to bypass the “Comments to the Author” section, enter your conflict of interest statement in the “Confidential to Editor” section, and submit your "Accept" recommendation.

Reviewer #2: All comments have been addressed

2. Is the manuscript technically sound, and do the data support the conclusions?

Reviewer #2: Yes

3. Has the statistical analysis been performed appropriately and rigorously? 

Reviewer #2: Yes

4. Have the authors made all data underlying the findings in their manuscript fully available?

Reviewer #2: Yes

5. Is the manuscript presented in an intelligible fashion and written in standard English?

Reviewer #2: Yes

6. Review Comments to the Author

Reviewer #2: Dear Authors,

Thank you for sharing the revised file of your manuscript and making the required changes based on the comments.

With my best wishes

7. PLOS authors have the option to publish the peer review history of their article (what does this mean?). If published, this will include your full peer review and any attached files.

Reviewer #2: No

---

## [Author Response · Author response to Decision Letter 1]

29 Jun 2022

A lot of medical research is based on observation and to avoid observational studies' reporting to be compromised for the quality, we recommend using STROBE checklist.

Please ensure the reporting of your observational study using the STROBE checklist (http://www.strobe-statement.org). The STROBE Statement consists of a checklist of 22 items, which relate to the title, abstract, introduction, methods, results, and discussion sections of articles.

Strobe checklist reviewed in line with manuscript.

Under Discussion, abbreviated terms 'TLEs and VLEs' are used without being spelled in full elsewhere. Please correct.

TEL and VLEs already spelled in full in the “Introduction” section see line 86 therefore manuscript not changed.

Title on online submission form and manuscript are identical. All duplicate files removed.

---

## [Editor Report · Decision Letter 2]

11 Jul 2022

Academic, clinical and personal experiences of undergraduate healthcare students during the COVID-19 pandemic: a prospective cohort study.

PONE-D-21-29579R2

Dear Dr. McFadden,

We’re pleased to inform you that your manuscript has been judged scientifically suitable for publication and will be formally accepted for publication once it meets all outstanding technical requirements.

Kind regards,

Kamran Sattar

Academic Editor

PLOS ONE
---

## [Editor Report · Acceptance letter]

18 Jul 2022

PONE-D-21-29579R2 

Academic, clinical and personal experiences of undergraduate healthcare students during the COVID-19 pandemic: a prospective cohort study. 

Dear Dr. McFadden:

I'm pleased to inform you that your manuscript has been deemed suitable for publication in PLOS ONE. Congratulations! Your manuscript is now with our production department. 

Kind regards, 

on behalf of

Dr. Kamran Sattar 

Academic Editor

PLOS ONE